# bFGF and SDF-1α Improve In Vivo Performance of VEGF-Incorporating Small-Diameter Vascular Grafts

**DOI:** 10.3390/ph14040302

**Published:** 2021-03-28

**Authors:** Larisa Antonova, Anton Kutikhin, Viktoriia Sevostianova, Elena Velikanova, Vera Matveeva, Tatiana Glushkova, Andrey Mironov, Evgeniya Krivkina, Amin Shabaev, Evgeniya Senokosova, Leonid Barbarash

**Affiliations:** Department of Experimental Medicine, Research Institute for Complex Issues of Cardiovascular Diseases, 6 Sosnovy Boulevard, 650002 Kemerovo, Russia; antolv@kemcardio.ru (L.A.); sevovv@kemcardio.ru (V.S.); veliea@kemcardio.ru (E.V.); matvvg@kemcardio.ru (V.M.); glushtv@kemcardio.ru (T.G.); miroav@kemcardio.ru (A.M.); kriveo@kemcardio.ru (E.K.); shabar@kemcardio.ru (A.S.); senoea@kemcardio.ru (E.S.); barbls@kemcardio.ru (L.B.)

**Keywords:** tissue engineering, regenerative medicine, tubular scaffolds, vascular grafts, two-layer structure, vascular endothelial growth factor, basic fibroblast growth factor, stromal cell-derived factor 1α, endothelial cells, smooth muscle cells

## Abstract

Tissue-engineered vascular grafts are widely tested as a promising substitute for both arterial bypass and replacement surgery. We previously demonstrated that incorporation of VEGF into electrospun tubular scaffolds from poly(3-hydroxybutyrate-co-3-hydroxyvalerate)/poly(ε-caprolactone) enhances formation of an endothelial cell monolayer. However, an overdose of VEGF can induce tumor-like vasculature; thereby, other bioactive factors are needed to support VEGF-driven endothelialization and successful recruitment of smooth muscle cells. Utilizing emulsion electrospinning, we fabricated one-layer vascular grafts with either VEGF, bFGF, or SDF-1α, and two-layer vascular grafts with VEGF incorporated into the inner layer and bFGF and SDF-1α incorporated into the outer layer with the following structural evaluation, tensile testing, and in vivo testing using a rat abdominal aorta replacement model. The latter graft prototype showed higher primary patency rate. We found that the two-layer structure improved surface topography and mechanical properties of the grafts. Further, the combination of bFGF, SDF-1α, and VEGF improved endothelialization compared with VEGF alone, while bFGF induced a rapid formation of a smooth muscle cell layer. Taken together, these findings show that the two-layer structure and incorporation of bFGF and SDF-1α into the vascular grafts in combination with VEGF provide a higher primary patency and therefore improved in vivo performance.

## 1. Introduction

Vascular tissue engineering has emerged as one of the most promising approaches for producing mechanically competent and biocompatible small-diameter vascular substitutes [1]. These constructs are fabricated of natural or synthetic biodegradable polymers to provide a scaffold for cell attachment, migration, and proliferation followed by de novo formation of the vascular tissue [2]. Ready-to-use, bioabsorbable, small-diameter vascular graft is still an unmet clinical need for both arterial bypass and arterial replacement [1].

We previously reported a spontaneous endothelialization of small-diameter electrospun vascular grafts blended of a natural crystalline aliphatic polyester poly(3-hydroxybutyrate-co-3-hydroxyvalerate) (PHBV) and a synthetic semi-crystalline aliphatic polyester poly(ε-caprolactone) (PCL) [3,4,5]. Either conjugation with arginine-glycine-aspartic acid (RGD) peptides or incorporation of vascular endothelial growth factor (VEGF) significantly and equally increased a primary patency rate of these grafts [5]. VEGF is frequently used to induce graft endothelialization since it promotes migration, proliferation, survival, and differentiation of endothelial cells (ECs), enhances nitric oxide production, and improves vascular permeability [6,7,8]. Although VEGF is considered as a potent growth factor to promote angiogenesis, its overdose, however, may lead to the development of immature, fenestrated capillaries similar to those in cancerous tissue [9].

Here we propose the incorporation of basic fibroblast growth factor (bFGF/FGF-2) and stromal cell-derived factor-1α (SDF-1α/CXCL12) to complement, support, and stabilize the effects of VEGF in vascular grafts. bFGF promotes migration, proliferation, and survival of ECs and smooth muscle cells (SMCs) [10]. Compared with VEGF, bFGF is a lesser pro-angiogenic factor, yet it promotes vessel maturation [11]. SDF-1α is another chemoattractant for ECs, stimulating formation of long and branched capillary networks [12]. Furthermore, SDF-1α enhances migration of bone-marrow-derived mesenchymal stromal cells which are able to differentiate into SMCs within the graft wall [13].

With the aim to test the effects of bFGF and SDF-1α in comparison with and in addition to VEGF, we implanted the grafts with incorporated VEGF, bFGF, SDF-1α, or all three factors into rat abdominal aortas for 3, 6, or 12 months with the following histological and immunohistochemical examination.

## 2. Results

### 2.1. Two-Layer Structure Allows the Separation of the Molecules with a Distinct Bioactivity

For the fabrication of the grafts, we used emulsion electrospinning, a well-established technique to provide a controllable and sustainable release and a synergistic delivery of bioactive agents [14,15]. This method allows chemical separation via the creation of an emulsion from: (1) an immiscible aqueous phase containing the dissolved bioactive compounds; (2) polymer solution dissolved in an organic phase, with the subsequent organization of the emulsified droplets into two distinct phases as the solvent evaporates from the electrospun fibers [16].

Grafts with incorporated VEGF, bFGF, or SDF-1α had a standard one-layer structure. To properly evaluate the possible synergistic action of these bioactive factors, we developed a graft containing VEGF incorporated into the inner layer and bFGF incorporated along with SDF-1α into the outer layer. Therefore, VEGF was reliably separated from bFGF and SDF-1α with respect to their different chemoattractant activity. This design conceived that VEGF might recruit progenitor and mature ECs to the luminal surface while bFGF and SDF-1α are capable of attracting cells, in particular SMCs, to produce extracellular matrix within the graft wall.

### 2.2. Two-Layer Structure Improves Structural and Tensile Properties of the Grafts

We first asked whether one-layer and two-layer grafts have similar structure and mechanical properties. Using scanning electron microscopy, we detected that all the grafts consisted of randomly distributed fibers and interconnected pores (Figure 1a). Notably, the two-layer grafts with all three bioactive factors had smaller fibers and pores compared to the one-layer grafts with either of the factors; however, the porosity did not change in both types of the grafts (Figure 1a,b). It has been reported that nanoscale fibers correlate with improved cell adhesion and proliferation compared to the fibers of larger diameter [17,18,19], possibly since micro- to nanoscale topography of electrospun fibers and pores better resemble the natural extracellular matrix [20,21,22]. None of the grafts had any visible defects. In addition, we did not detect any clear boundary between the layers in the two-layer grafts that is crucial for the prevention of spontaneous delamination upon the implantation into the vasculature.

To withstand and properly adapt to the blood pressure upon implantation into the arterial vasculature, the grafts should have high tensile strength and elasticity as similar to the native arteries as possible [23]. For the comparison with our polymer grafts, we selected internal mammary artery since it is widely applied as an autologous graft for bypass surgery [24]. A mechanical testing showed certain differences between the grafts, with the higher tensile strength and elastic properties in two-layer grafts with all three bioactive factors incorporated (Figure 2). Two-layer grafts were closer to internal mammary artery in terms of tensile elasticity and tensile elongation than one-layer scaffolds (Figure 2).

Taken together, these results indicate that the two-layer structure confers enhanced structural and tensile properties in comparison with the one-layer structure.

### 2.3. Incorporated Bioactive Factors Undergo Controlled Release and Retain Their Functional Activity

To ensure the constant and high rate of vascular regeneration, bioactive factors incorporated into the polymer fibers should be released into the microenvironment in a sustainable and controllable manner, although an initial burst is unavoidable regardless of an electrospinning technique [6,14,15,16,20,21]. Yet, emulsion electrospinning permits alleviation of this burst, retaining higher amounts of the bioactive factors within the fibers for the subsequent prolonged release and thereby better protecting them from the degradation [20]. Another critical prerequisite for the regulated release is a low and stable biodegradation rate of the electrospun fibers under physiological conditions [6,14,15,16,20,21]. We therefore incubated the cross-sectioned polymer scaffolds in vitro at 37 °C, 95% air: 5% CO_2_ and humidified environment and measured the kinetics of VEGF release over time. Upon the initial burst (≈1700 pg released during the first 2 days), we documented a steady increment of VEGF in the milieu indicative of its gradual release (Figure 3a). Over 80 days, cumulative amount of VEGF released into the milieu was ≈5000 pg (out of 50,000 pg incorporated into the scaffold), suggesting a slow degradation of the PHBV/PCL scaffold at physiological conditions (Figure 3a).

Next, we investigated the temporal changes in quantities of circulating bioactive factors upon the implantation of vascular grafts containing VEGF, bFGF, SDF-1α, or all factors combined, into the abdominal aortas of Wistar rats. Serum VEGF, but not bFGF or SDF-1α, showed a notable increase 3 months postimplantation in both respective groups (VEGF- and VEGF/bFGF/SDF-1α-incorporating grafts, Figure 3b–d). Yet, two-layer grafts with combined growth factors exhibited a superior profile of their release, as VEGF was also elevated in the serum of these animals 12 months postimplantation and bFGF was also augmented over the study period, yet gradually decreasing to the 12th month (Figure 3b–d). 

In keeping with in vitro kinetics of VEGF release and quantitative measurements of the incorporated growth factors in the rat serum, implanted grafts consistently demonstrated a low biodegradation rate at each time point, remaining intact upon the excision (Figure 4a,b).

Besides the defined release of incorporated molecules, they must retain their functional activity to guide the repopulation of the graft wall with the host cells, which both enhance enzymatic degradation of the polymer fibers and synthesize the extracellular matrix to reproduce the native blood vessel [23]. As VEGF is a major pro-angiogenic factor leading to the evident neovascularization upon its administration, we performed a gross examination of the explanted grafts 3, 6, and 12 months postimplantation utilizing the freshly implanted scaffolds as a negative control. Strikingly, a vasa vasorum meshwork was notable as early as 3 months postimplantation in VEGF- and VEGF/bFGF/SDF-1α-incorporating grafts, while grafts without VEGF were devoid of major neovessels (Figure 4b). Taken together, these observations suggested a retained functional activity of polymer-incorporated VEGF (and similar small molecules such as bFGF and SDF-1α) in vivo.

### 2.4. Grafts with Incorporated VEGF, bFGF, and SDF-1α Have an Increased Primary Patency Rate

We then evaluated the biological effects of bFGF and SDF-1α to examine their possible synergistic action with VEGF. Rat aorta replacement model is considered of particular utility for vascular tissue engineering since it permits high-powered and cost-effective experimentation and provides abundant tissue to examine specific molecular markers [25,26]. Hence, we implanted the grafts modified with either VEGF, bFGF, SDF-1α, or all three factors into abdominal aortas of Wistar rats. Animals were sacrificed 3, 6, or 12 months postimplantation, with the subsequent collection of the grafts for the analysis. As an excessive transanastomosal ingrowth of cells is a characteristic of rat but not human vasculature [27], we assessed only midgraft area but not anastomoses.

As a favorable outcome of implantation, we selected primary patency, a desirable endpoint in both experimental and clinical vascular surgery. To assess the primary patency rate, we performed hematoxylin and eosin (H&E) staining. Intriguingly, 3 months postimplantation all the graft types demonstrated an equal primary patency rate of 75% (3 out of 4 implanted grafts), whereas 6 and 12 months postimplantation it decreased to 50% (2 out of 4) for bFGF- and SDF-1α-modified grafts but reached 100% (4 out of 4) for the grafts with all three factors incorporated (Figure 5a,b).

### 2.5. bFGF and SDF-1α Do Not Affect the Number of Collagen-Producing Cells within the Grafts

To explain an increased primary patency, we suggested three potential physiological mechanisms contributing to arterial homeostasis: (1) an enhanced migration and/or higher viability of the collagen-producing cells within the grafts since collagen content is a crucial factor providing arterial integrity [28,29]; (2) an improved endothelialization of the luminal surface, which has been previously recognized as a key factor for preventing thrombosis and therefore successful long-term rat abdominal aorta replacement [5]; (3) an accelerated formation of a SMC layer responsible for the proper contractility of a vascular wall [30,31,32,33]. 

We first investigated whether either bFGF or SDF-1α affect the number of collagen-producing cells within the grafts compared to VEGF. To evaluate this, we performed a double immunostaining for collagens I and IV, which are major collagen types in the arterial wall [28,29]. Expectedly, we identified abundant collagen fibers across all the grafts but did not find any differences in the number of collagen-producing cells between almost all types of the grafts at all the time points (Figure 6a,b).

### 2.6. bFGF and SDF-1α Support VEGF-Induced Endothelialization of the Luminal Surface

Further, we hypothesized that either bFGF or SDF-1α can recruit specific cell populations or increase their viability instead of raising total number of functional cells. Hence, we tested the possible beneficial effects of bFGF or SDF-1α on endothelialization of the luminal surface. We first stained the grafts for von Willebrand factor (vWF), which is an EC-secreted glycoprotein establishing contacts between collagens, laminins and ECs [34]. Either bFGF or SDF-1α did not significantly induce endothelialization in comparison with VEGF at all the time points; however, vWF-producing ECs were identified in all the grafts (Figure 7a,b). 

We then suggested that both of these factors can support VEGF-induced endothelialization being unable to induce a significant endothelialization per se. To examine this hypothesis and to further explore the assumed functional endothelium, we performed a double immunostaining using specific mature endothelial cell marker CD31 and endothelial progenitor cell marker CD34 [35,36]. Notably, we found that incorporation of bFGF and SDF-1α in addition to VEGF led to the formation of an EC monolayer at the luminal surface 6 and 12 months postimplantation but this was not the case for bFGF or SDF-1α added alone (Figure 8a,b). Expectedly, no specific effects of incorporated bioactive factors were revealed with respect to CD34-positive cells (Figure 8a,b).

### 2.7. bFGF Facilitates Formation of a Smooth Muscle Cell Layer

Finally, we carried out an α-smooth muscle actin (α-SMA) staining to detect SMCs within the grafts. Strikingly, we identified an increased number of α-SMA-positive cells forming the organized tissue layers within the grafts with bFGF and all three bioactive factors as early as 3 months postimplantation (Figure 9a,b). Hence, we suggested bFGF as an enhancer of SMC layer formation. Noteworthy, the dense layers of a smooth muscle tissue were detected in all the grafts 12 months postimplantation (Figure 9a,b).

## 3. Discussion

Monolayer of ECs, SMC layer, and abundant organized fibers of collagen, elastin, and other extracellular matrix proteins in tissue-engineered vascular prostheses resemble the structure of a native artery [28,29,30,31,32,33]. It is widely accepted that an early-formed EC monolayer prevents thrombosis while a SMC layer and organized structure of the extracellular matrix with a sufficient amount of collagen and elastin provide contractility and compliance close to native arteries [6,30,31,32,33,37,38]. Hence, rapid endothelialization, formation of a SMC layer, and a high number of functional cells producing extracellular matrix proteins are all considered as the crucial factors providing a high primary patency rate [6,30,31,32,33,37,38]. Incorporation of the growth factors into electrospun polymer scaffolds is a widely established strategy to promote cell adhesion, increase proliferation, improve cell viability, and enhance cellular infiltration of the vascular grafts [6,20,39,40]. Among the variety of available bioactive factors, we selected VEGF, bFGF, and SDF-1α due to their specific abilities to recruit and sustain ECs and SMCs [6,7,8,10,11,12,13]. 

Here we demonstrated that both bFGF and SDF-1α were not active recruiters of collagen-producing cells or major inducers of endothelialization per se compared to VEGF; instead, they supported VEGF-induced endothelialization of the luminal surface. Furthermore, bFGF promoted formation of a SMC layer. Together with improved structural and tensile properties, these two effects resulted in a higher primary patency rate of the two-layer grafts with VEGF, bFGF, and SDF-1α in comparison with the one-layer grafts with either of these factors. 

Importantly, both in vitro and in vivo experiments attested a low biodegradation rate of the PHBV/PCL vascular grafts, suggesting a sustained and controlled release of the incorporated bioactive factors. Vasa vasorum network revealed exclusively in the VEGF-incorporating grafts testified to the functional activity of these factors. Conceivably, the functional consequences of VEGF, bFGF, and SDF-1α activity are delayed and stretched out in time due to the low biodegradation rate of the polymer blend and relatively slow vascular regeneration. Functional activity of VEGF, bFGF, and SDF-1α 3 months postimplantation could result in respective changes at the following time points, as the graft wall is a multi-component and dynamic microenvironment having complex spatiotemporal patterns of evolution.

To the best of our knowledge, bFGF was not tested by an orthotopic in vivo implantation as an agent enhancing endothelialization and SMC layer formation in tissue-engineered vascular grafts. Strikingly, it has been shown that SDF-1α immobilized onto heparin conjugated to the microfibrous vascular grafts recruited endothelial progenitor cells to the luminal surface and smooth muscle progenitor cells to the graft wall, with their following differentiation into ECs and SMCs, respectively [41]. Moreover, SDF-1α-immobilized grafts had higher elastic modulus [41]. Taken together, these effects improved long-term patency [41]. Another study demonstrated a capability of SDF-1α to reduce inflammatory signaling within the bioactivated scaffolds both in vitro and in rat abdominal aorta replacement model [42]. Our results are consistent with these studies despite the distinct techniques of SDF-1α application, i.e., incorporation into the graft, immobilization at its luminal surface [41], and preparation of the synthetic SDF-1α-derived peptides [42]. Furthermore, results from in vitro or non-orthotopic implantation studies also confirmed an ability of either bFGF or SDF-1α to promote migration of human ECs, mesenchymal progenitor cells, and myofibroblasts to the various scaffolds [43,44,45]. 

To sum up, two-layer grafts fabricated employing emulsion electrospinning with VEGF incorporated into the inner layer and bFGF along with SDF-1α incorporated into the outer layer rapidly acquire a monolayer of ECs and a SMC layer providing a high primary patency rate. Hence, they can be suggested as the promising candidates for a preclinical testing on a senescent sheep common carotid artery replacement model where the anatomic, hemodynamic, and coagulation conditions are similar to the human vasculature [26,46,47].

## 4. Materials and Methods

### 4.1. Graft Preparation

Small-diameter vascular grafts were fabricated using emulsion electrospinning (Nanon-01A, MECC, Tokyo, Japan) from PHBV (403105, Sigma-Aldrich, Saint Louis, MO, USA):PCL (440744, Sigma-Aldrich, Saint Louis, MO, USA) (5%:10%)/chloroform (366927, Sigma-Aldrich, Saint Louis, MO, USA) solution using the following parameters: 23 kV voltage, 0.5 mL/h feed rate, 2 mm rotating drum diameter, 22G needle, and 150 mm tip-to-collector distance. Either VEGF (SRP4365, Sigma-Aldrich, St. Louis, MO, USA), bFGF (SRP4039, Sigma-Aldrich, St. Louis, MO, USA), or SDF-1α (SRP3252, Sigma-Aldrich, St. Louis, MO, USA) were dissolved in phosphate buffered saline (10010023, Thermo Fisher Scientific, Waltman, MA, USA) to 10 µg/mL concentration and then added to PHBV/PCL/chloroform solution (1:20), with the final concentration of 500 ng/mL. Grafts with the combination of VEGF, bFGF, and SDF-1α were two-layered, with the inner layer fabricated using 27G needle and containing VEGF (500 ng/mL) and the outer layer prepared utilizing 22G needle and containing bFGF and SDF-1α (500 ng/mL each).

### 4.2. Morphological Assessment

Qualitative and quantitative assessment of the structural properties was performed using scanning electron microscopy and quantitative image analysis, respectively, as in our previous study [4]. 

### 4.3. Mechanical Testing

Assessment of tensile properties (6 grafts per group) was performed using uniaxial tension test employing universal testing machine (Zwick/Roell, Ulm, Germany) in a longitudinal direction with a working segment length of 1 cm, preload of 0.01 N, nominal force of 50 N, and crosshead speed of 10 mm/min. Sample thickness was measured utilizing a feeler gauge (Krisbow, Jakarta, Indonesia). We evaluated yield and ultimate tensile strength, tensile elasticity, and tensile elongation with the following calculation of the stress–strain curve. Internal mammary artery samples (*n* = 6) were used as the controls upon collection from the patients who underwent coronary artery bypass graft surgery.

### 4.4. In Vitro Kinetics of VEGF Release

To evaluate VEGF release from polymer fibers during their degradation, we incubated ethanol-treated and thoroughly washed VEGF-containing scaffolds (1 × 2 cm^2^ segments) in a phosphate-buffered saline (1.5 mL tubes) at 37 °C, 95% air: 5% CO_2_ atmosphere and high humidity. The milieu (200 µL) was sequentially collected at 2, 4, 8, 12, 16, 20, 24, 28, 32, 40, 48, 64, and 80th day of incubation with the following cumulative quantification of VEGF concentration by means of enzyme-linked immunosorbent assay (RRV00; R&D Systems, Minneapolis, MN, USA). At each time point, withdrawn volume was refilled by the sterile phosphate-buffered saline.

### 4.5. In Vivo Implantation

For the animal experiments, we used Wistar rats (male, 6-month-old, 400–450 g body weight, *n* = 48). Ethylene oxide-sterilized 10 mm length and 2 mm diameter grafts with incorporated VEGF, bFGF, SDF-1α, or all three bioactive factors (*n* = 12 per group) were implanted into abdominal aorta. Animal handling and surgery were performed as in [4]. One-third (*n* = 4) of rats in each group was sacrificed 3, 6, or 12 months postimplantation by an intraperitoneal injection of a sodium pentobarbital (100 mg/kg body weight, P-010, Sigma-Aldrich, St. Louis, MO, USA). Equal parts of the explanted grafts and adjacent aortic tissue were either frozen at −140 °C or fixed in 10% phosphate buffered formalin (06-001, BioVitrum, St. Petersburg, Russia) as in [4].

### 4.6. In Vivo Kinetics and Functional Analysis of Bioactive Factors

To investigate in vivo kinetics of the bioactive factors incorporated into the grafts, we collected the serum from the experimental animals (*n* = 4 per group) at each time point (baseline, 3, 6, and 12 months) with the subsequent quantification of VEGF (RRV00, R&D Systems, Minneapolis, MN, USA), bFGF (MFB00, R&D Systems, Minneapolis, MN, USA), and SDF-1α (SEA122Ra, Cloud-Clone, Houston, TX, USA) using enzyme-linked immunosorbent assay. Functional activity of the bioactive factors during the implantation was assessed at gross examination of the excised grafts by the presence of visible neovessels indicative of a *vasa vasorum* network.

### 4.7. Histological Examination

After being fixed with formalin, the grafts were embedded in paraffin (10342, Histomix Extra, BioVitrum. St. Petersburg, Russia), further stained with hematoxylin and eosin (ab245880, Abcam, Cambridge, UK), and finally examined by light microscopy (Axio Imager A1, Carl Zeiss, Oberkochen, Germany) as in [4].

### 4.8. Immunofluorescence Examination

Snap-frozen tissue samples were sectioned, treated, and stained as in our previous studies [4,5]. Native rat aorta and bovine serum albumin (A3059, Sigma-Aldrich, St. Louis, MO, USA) solution were used as a positive and negative control, respectively. Slides were examined by confocal laser scanning microscopy using LSM 700 (Carl Zeiss, Oberkochen, Germany) as in [4,5]. Quantitative image analysis was conducted utilizing ImageJ software (National Institutes of Health, Bethesda, MD, USA) as in [5], with 6 images evaluated for each group per each time point. Exceptionally for α-SMA staining, we counted not the number of positive cells but the percent of positively stained area.

### 4.9. Immunohistochemistry

The protocol of α-SMA staining was the same as in [4] excepting the use of respective primary antibody from Spring Bioscience (rabbit anti-α-SMA, M4710, Spring Bioscience, Pleasanton, CA, USA). Sections were examined by light microscopy (Axio Imager A1, Carl Zeiss, Oberkochen, Germany).

### 4.10. Statistical Analysis

Statistical analysis was carried out utilizing GraphPad Prism (GraphPad Software, San Diego, CA, USA). As there were not enough data for assessing the sample distribution, we considered all the data as not normally distributed. Descriptive data were represented by the median solely or the median with interquartile range. Kruskal–Wallis test was applied to compare the groups, with the further adjustment for multiple comparisons using Dunn’s multiple comparisons test. Adjusted *p*-values ≤ 0.05 were defined as significant.

## 5. Conclusions

Two-layer vascular grafts with VEGF incorporated into the inner layer and bFGF and SDF-1α incorporated into the outer layer have improved endothelialization and better in vivo performance as compared to one-layer grafts with either of these bioactive factors.

## Figures and Tables

**Figure 1 pharmaceuticals-14-00302-f001:**
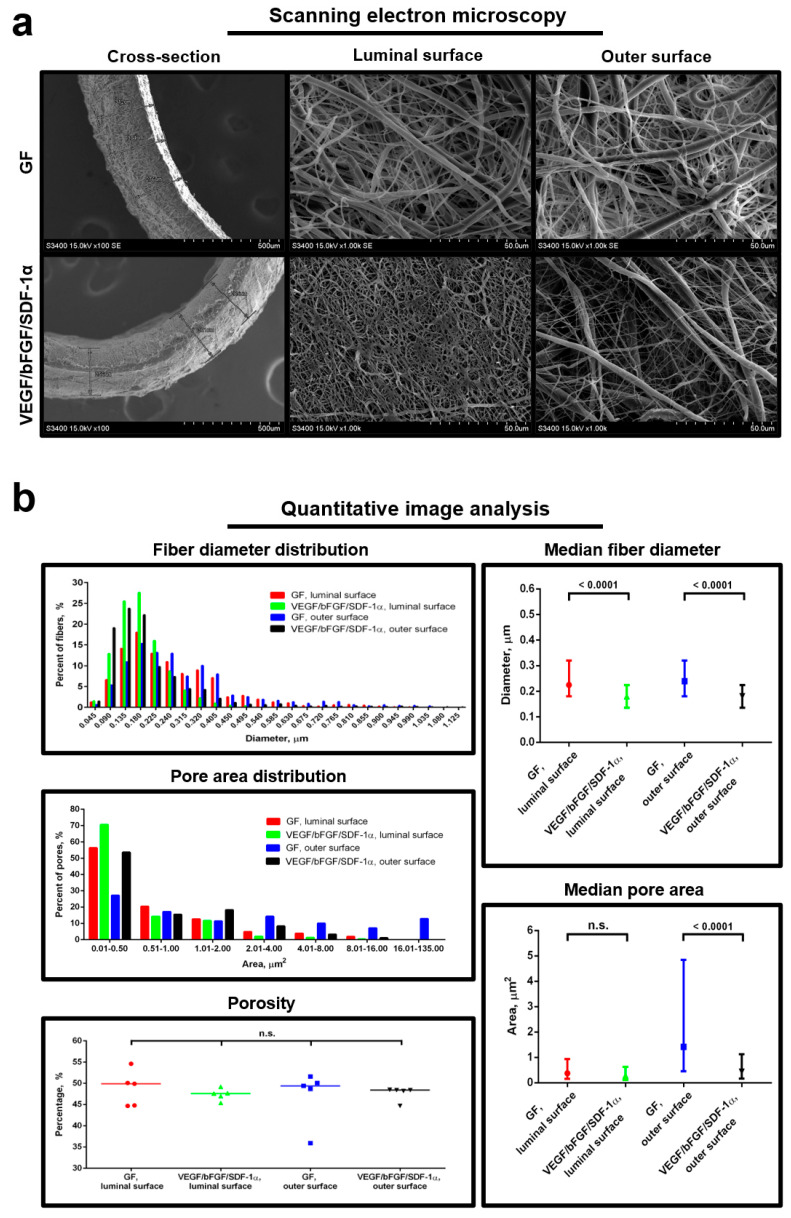
Two-layer structure of the grafts scales down the topography of the fibers and pores. (**a**) Pre-implantation representative scanning electron microscopy images of the grafts modified exclusively with either VEGF, bFGF or SDF-1α (merged in GF group due to being one-layer), or with all three factors having a two-layer structure; (**b**) Quantitative image analysis (six images per group containing in total > 100 fibers and pores), data are represented as the median with interquartile range, Kruskal–Wallis test with the further Dunn’s multiple comparisons test, *p* values adjusted for multiple comparisons are reported in a numerical manner, n.s. is for not significant.

**Figure 2 pharmaceuticals-14-00302-f002:**
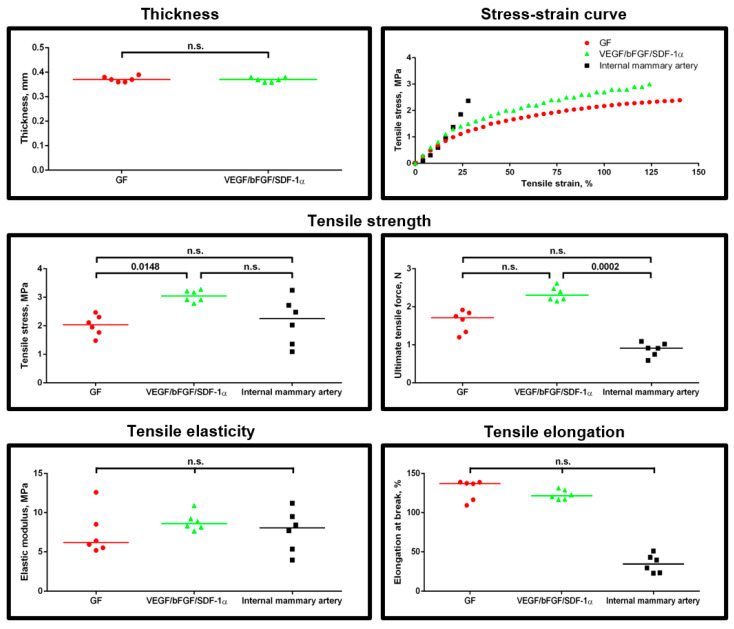
Two-layer structure endows the grafts with improved mechanical properties. Mechanical testing of the grafts modified exclusively with either VEGF, bFGF or SDF-1α (merged in GF group due to being one-layer), or with all three factors having a two-layer structure. Data are represented as a univariate scatterplot with the median, each dot represents a measurement from one graft, Kruskal–Wallis test with the further Dunn’s multiple comparisons test, *p* values adjusted for multiple comparisons are reported in a numerical manner, n.s. is for not significant. For a stress-strain curve, each dot represents the median of respective measurements for all six graft samples from the testing.

**Figure 3 pharmaceuticals-14-00302-f003:**
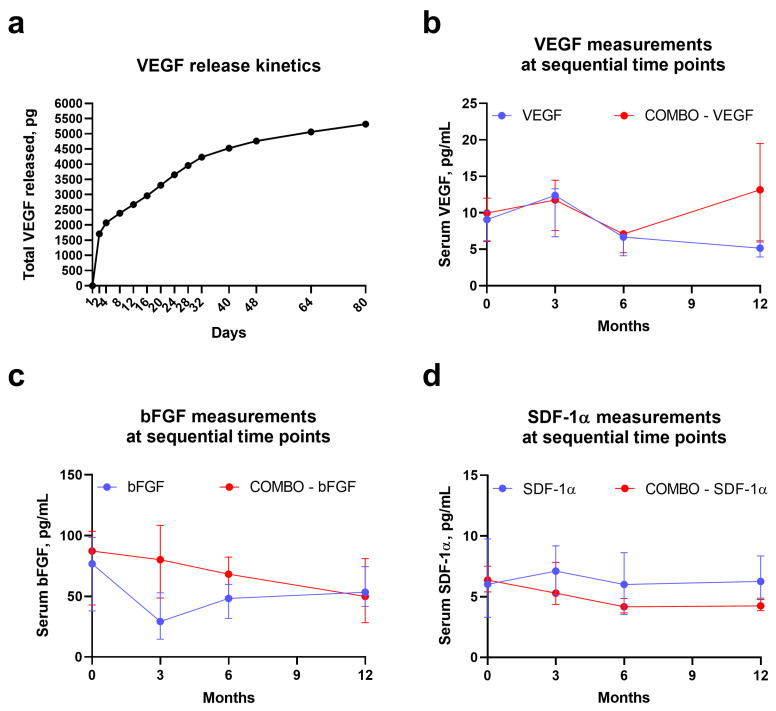
PHBV/PCL vascular grafts ensure sustained and controlled release of bioactive factors. (**a**) Evaluation of in vitro kinetics of VEGF release by incubation of the scaffolds in phosphate-buffered saline and subsequent enzyme-linked immunosorbent assay of the sequentially collected milieu; Quantitative analysis of temporal changes in VEGF (**b**), bFGF (**c**), and SDF-1α (**d**) in the serum of Wistar rats which underwent the implantation of the grafts into the abdominal aorta for the indicated time points. Note an increase in VEGF upon 3 months of the implantation and superior kinetics profile of two-layer grafts with all three bioactive factors incorporated. Blue dots and lines indicate the values from the animals with the grafts containing VEGF, bFGF, or SDF-1α alone. Red dots and lines indicate the values from the animals which underwent implantation of the grafts with all three bioactive factors combined (COMBO).

**Figure 4 pharmaceuticals-14-00302-f004:**
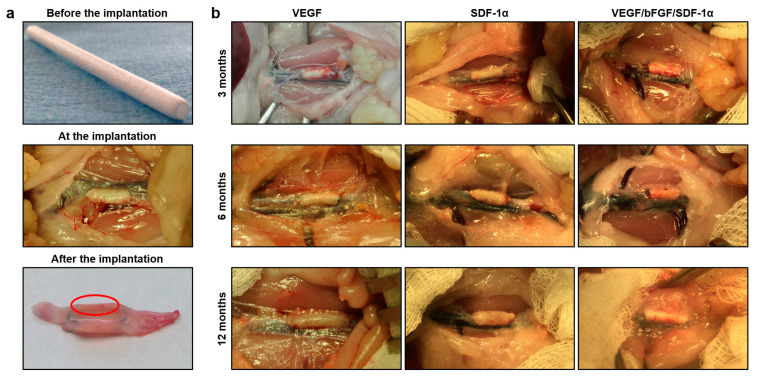
Gross examination of the vascular grafts. (**a**) Electrospun scaffold before the implantation, at the implantation into the rat abdominal aorta and its cross-section after the explantation. Red circle demarcates intramural vasa vasorum; (**b**) One-layer grafts containing exclusively VEGF or SDF-1α or two-layer grafts with all three bioactive factors (VEGF, bFGF, and SDF-1α) implanted into the rat abdominal aorta. Note the integrity of the polymer scaffold and notable vasa vasorum network in VEGF/bFGF/SDF-1α-containing grafts at all the time points.

**Figure 5 pharmaceuticals-14-00302-f005:**
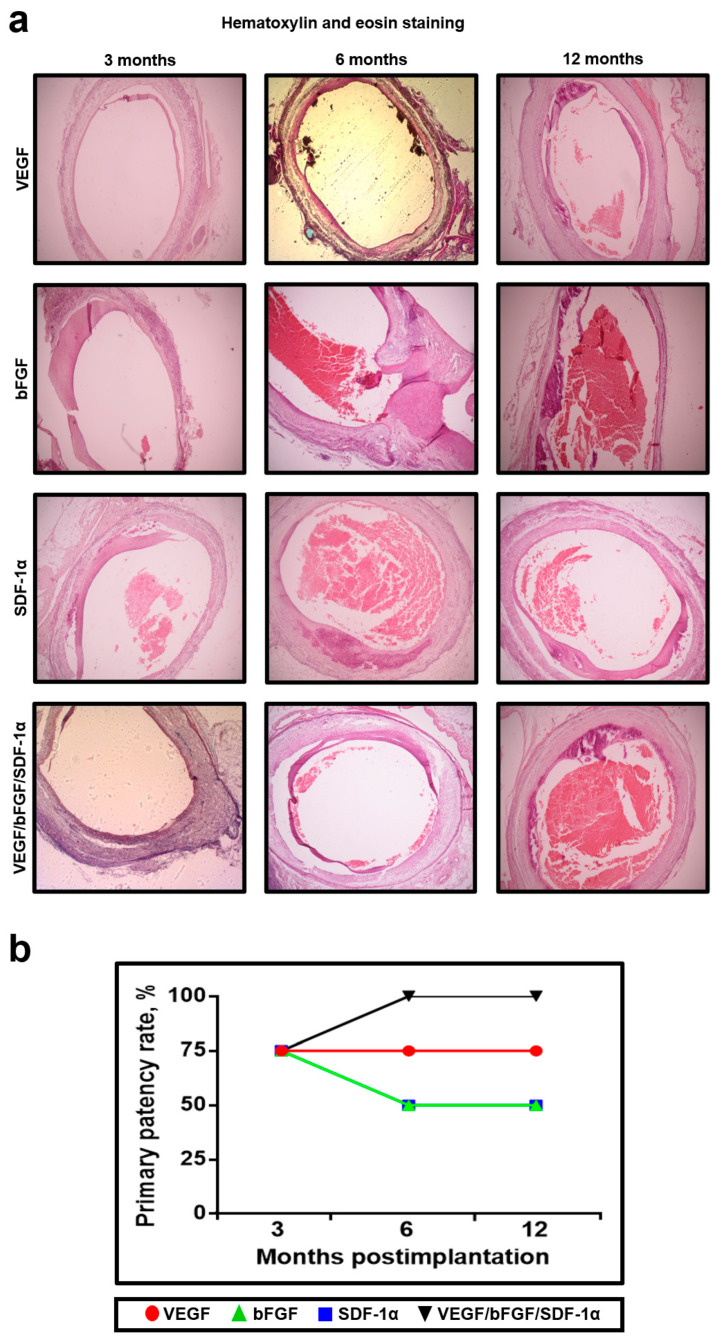
Grafts modified with a combination of VEGF, bFGF, and SDF-1α have a higher long-term primary patency rate than the grafts with either of these factors. (**a**) Representative images of the H&E-stained grafts modified with either VEGF, bFGF, SDF-1α, or all three factors at 3, 6 and 12 months postimplantation, ×50 magnification; (**b**) Quantitative image analysis, each dot represents a primary patency rate with respect to the experimental group and time point.

**Figure 6 pharmaceuticals-14-00302-f006:**
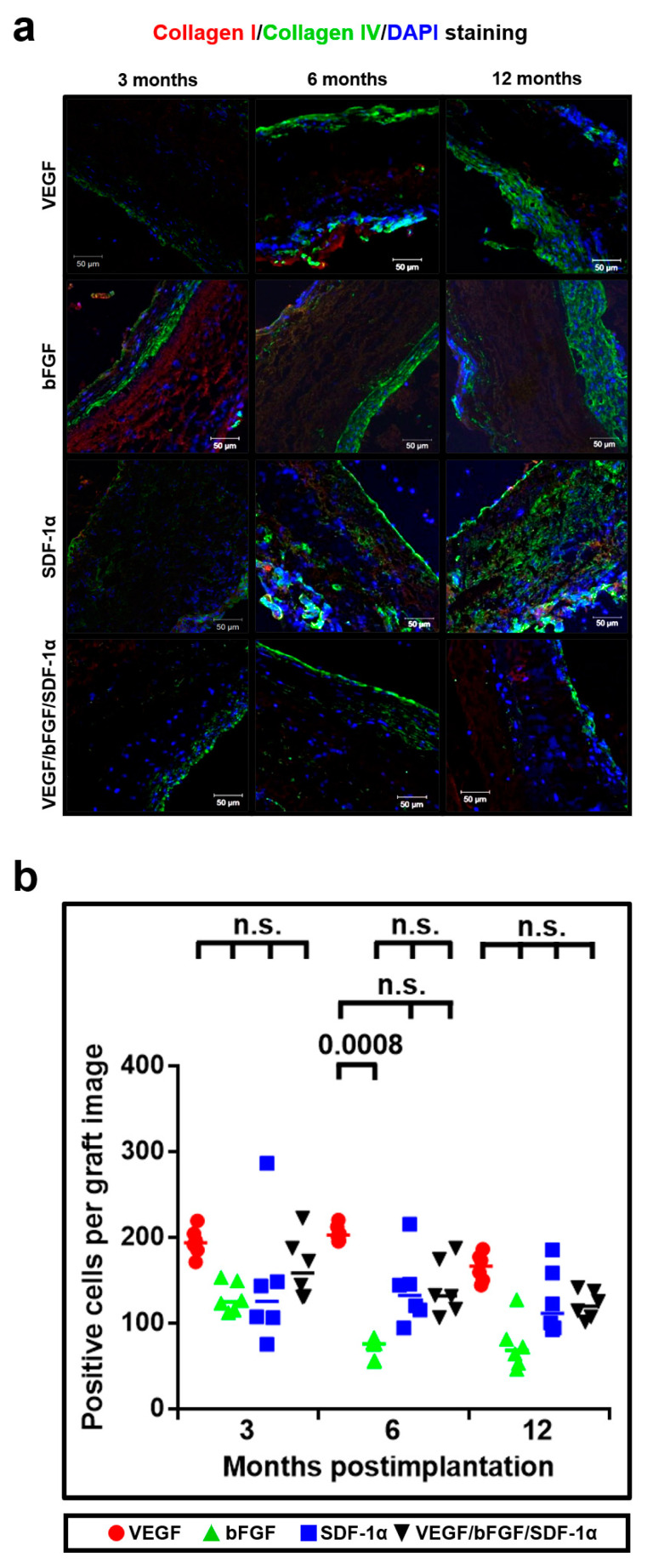
Either bFGF or SDF-1α do not change the number of collagen-producing cells within the grafts. (**a**) Representative images of collagen I- (red), collagen IV- (green), and DAPI- (blue) stained grafts modified with either VEGF, bFGF, SDF-1α, or all three factors at 3, 6 and 12 months postimplantation; (**b**) Quantitative image analysis, data are represented as a univariate scatterplot with the median, each dot represents a count from one image, Kruskal–Wallis test with the further Dunn’s multiple comparisons test, *p* values adjusted for multiple comparisons are reported in a numerical manner, n.s. is for not significant.

**Figure 7 pharmaceuticals-14-00302-f007:**
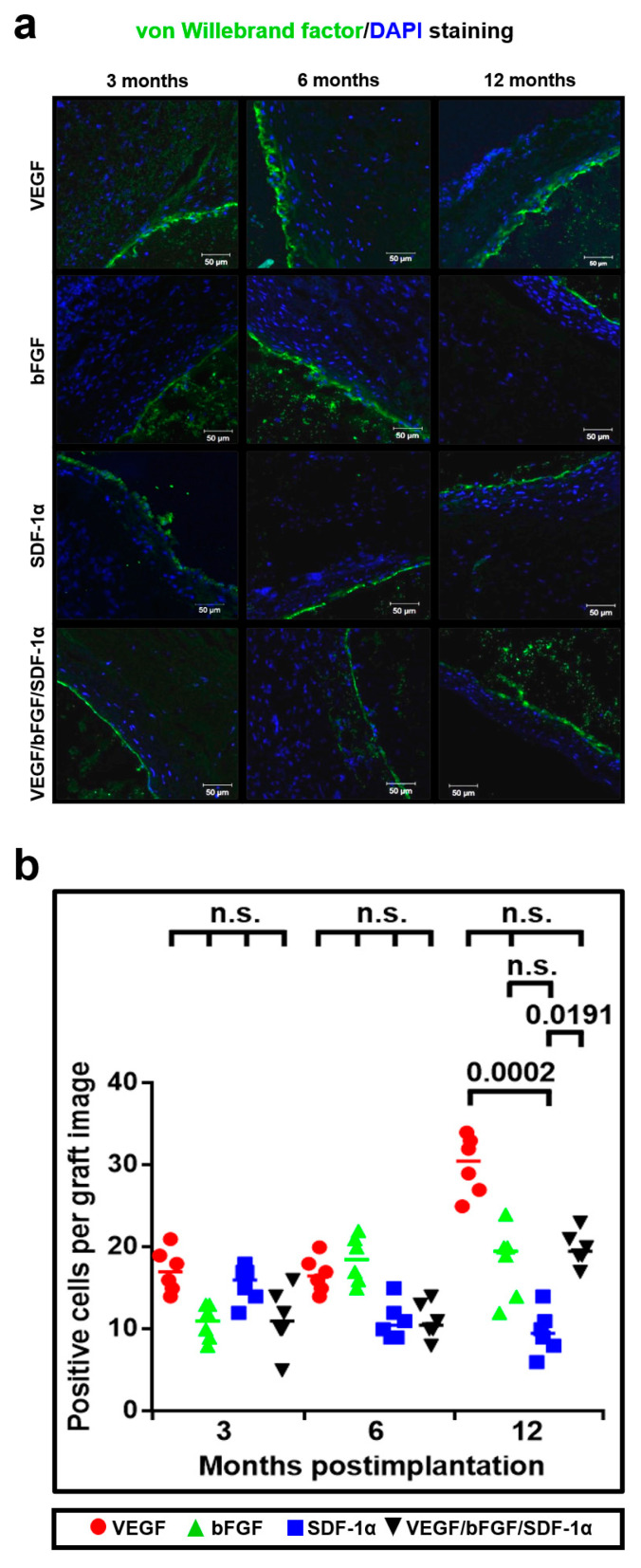
Either bFGF or SDF-1α do not induce significant endothelialization of the luminal surface compared with VEGF. (**a**) Representative images of vWF- (green) and DAPI- (blue) stained grafts modified with either VEGF, bFGF, SDF-1α, or all three factors at 3, 6 and 12 months postimplantation; (**b**) Quantitative image analysis, data are represented as a univariate scatterplot with the median, each dot represents a count from one image, Kruskal–Wallis test with the further Dunn’s multiple comparisons test, *p* values adjusted for multiple comparisons are reported in a numerical manner, n.s. is for not significant.

**Figure 8 pharmaceuticals-14-00302-f008:**
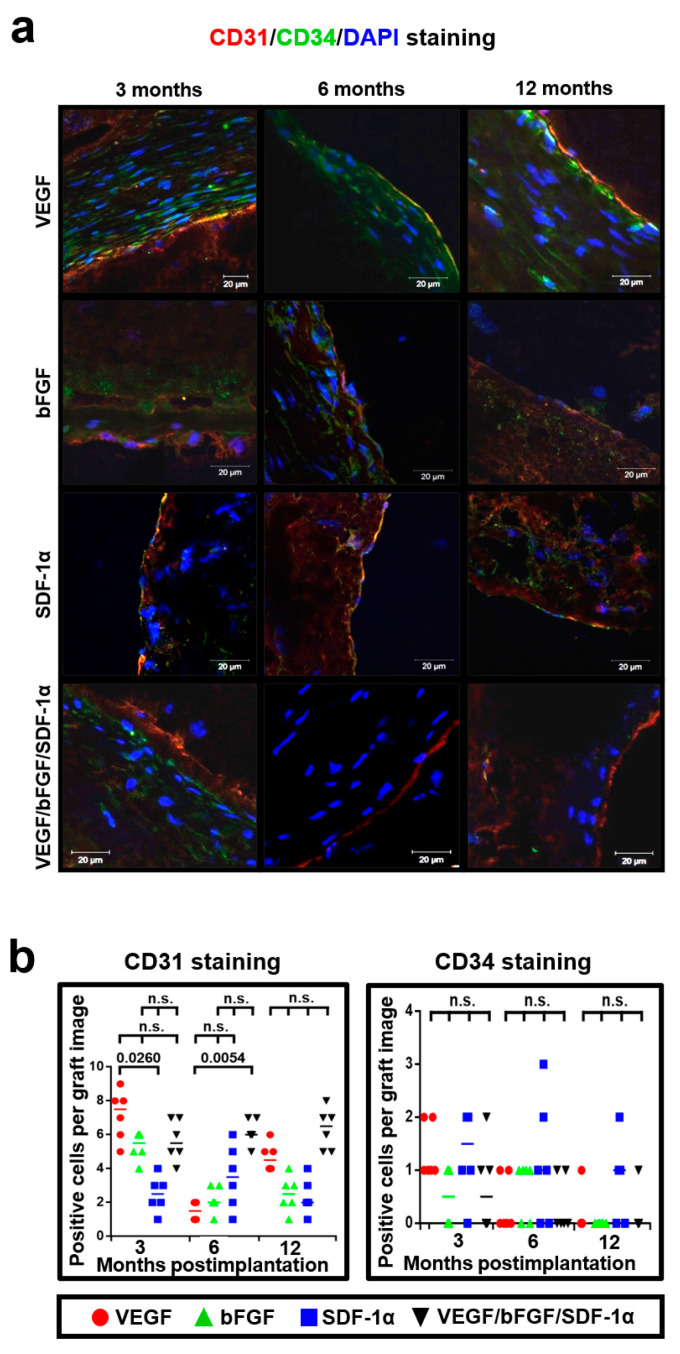
bFGF and SDF-1α support VEGF-induced endothelialization of the luminal surface. (**a**) Representative images of CD31- (red), CD34- (green), and DAPI- (blue) stained grafts modified with either VEGF, bFGF, SDF-1α, or all three factors at 3, 6 and 12 months postimplantation; (**b**) Quantitative image analysis, data are represented as a univariate scatterplot with the median, each dot represents a count from one image, Kruskal–Wallis test with the further Dunn’s multiple comparisons test, *p* values adjusted for multiple comparisons are reported in a numerical manner, n.s. is for not significant.

**Figure 9 pharmaceuticals-14-00302-f009:**
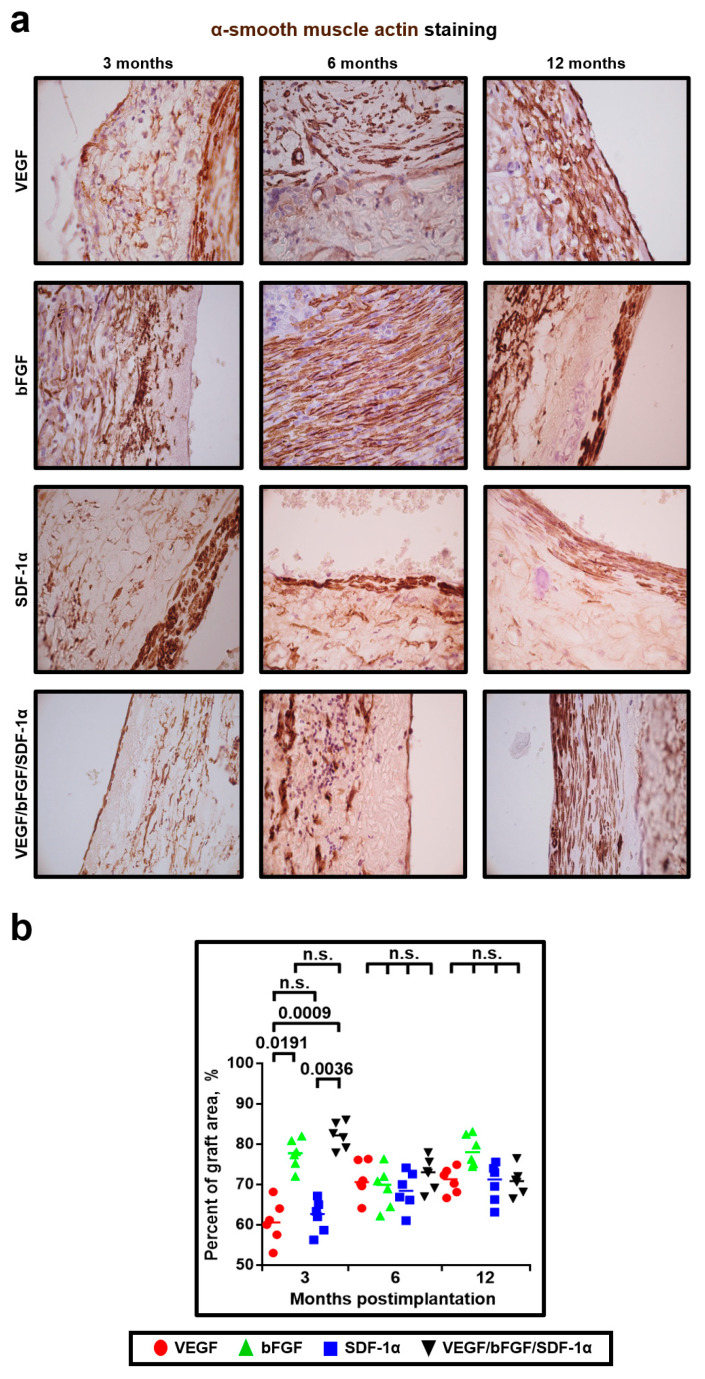
bFGF accelerates formation of a smooth muscle cell layer. (**a**) Representative images of α-SMA- (brown) stained grafts modified with either VEGF, bFGF, SDF-1α, or all three factors at 3, 6 and 12 months postimplantation; (**b**) Quantitative image analysis, data are represented as a univariate scatterplot with the median, each dot represents a count from one image, Kruskal–Wallis test with the further Dunn’s multiple comparisons test, *p* values adjusted for multiple comparisons are reported in a numerical manner, n.s. is for not significant.

## Data Availability

The datasets generated during and/or analysed during the current study are available from the corresponding author on reasonable request.

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
