# Peer review of "bFGF and SDF-1α Improve In Vivo Performance of VEGF-Incorporating Small-Diameter Vascular Grafts"

_pharmaceuticals, 2021, doi:10.3390/ph14040302_

Round 1

Reviewer 1 Report

The manuscript by Antonova et al. described the growth factor loaded fibers electrospun from poly(3-hydroxybutyrate-co-3-hydroxyvalerate)/poly(ε-caprolactone). The manuscript was well written and the research setup was appropriate.

A minor suggestion to the authors is that the graphical illustrations (e.g., Figure 1 - Figure 5) were very hard to read due to the small font size. Please reorganize the presentation style to make the readers easy on the data.

Author Response

We sincerely thank the reviewer for the high evaluation of our study. Please see the attachment. 

Reviewer 2 Report

n the current manuscript Antonova et al study the effect of incorporating VEGF, bFGF  and/or SFD1 into electrospun tissue engineered blood vessels with regard to re-endothelialisation in vivo in rats. They fabricated one-layer vascular grafts with either VEGF, bFGF, or SDF-1α, and  two-layer vascular grafts with VEGF incorporated into the inner layer and bFGF and SDF-1α  incorporated into the outer layer with the following structural evaluation, tensile testing, and in vivo testing using a rat abdominal aorta replacement model for 3, 6 or 12 months, with only 4 rats per group.

The manuscript is well written, but mainly focused on the technically very interesting electrospinning techniques, and not that much on the biological mechanisms involved. Simply reporting the encouraging long-term biological effects on graft patency in not sufficient, nor the staining for cells do collagen.

Crucial control experiments should be added, e.g. to demonstrate that the incorporated VEGF, bFGF and SFD-1 are actually still biologically active in the grafts.

Moreover, this becomes especially important after long-term in vivo positioning since the authors implanted the grafts with incorporated VEGF, bFGF, SDF-1α, or all three factors into rat abdominal aortas for 3, 6, or 12 months with the following histological and immunohistochemical examination. It is difficult to believe that in the incorporated factors are active for such a long period.

This raises a series of questions that need to be addressed. For how long is are the incorporated factors present in the grafts? Are they released? Or degraded? Are they functionally active in vivo? 

And the key question related to these questions and the main findings of the manuscript: How do the authors explain that the beneficial effects of the triple factor inclusion in the grafts on graft patency only becomes detectable after 6 and 12 months, and no effects are seen after three months ? It is difficult to believe the factors are still active after 6-12 months in vivo engraftment of the material.

Line 78—81: “In our model VEGF recruits progenitor and mature ECs to the luminal surface while bFGF and SDF-1α attract functional cells, in particular SMCs, which further produce glycosaminoglycans, collagens, elastin, and other extracellular matrix (ECM) proteins within the graft wall”, where are the experimental data supporting this remark?

Fig3 A , B. In the panel in the lower right corner, 12 months , combined incubation with VEGF, bFGF, SDF-1α, the graft shows a profound occluding lesion, yet a 100% patency rate is claimed for this group in panel B after 12 months. How can this be explained?

Lines 146 , section 2.4: it is difficult to understand why these effects only become detectable after more than 6 months of engraftment. That should be studied in more details, e.g. using multiple time point analysis.

Line 172: they tested the possible beneficial effects of bFGF or SDF-1α on endothelialization of the luminal surface, but at what time point? as said above, it is difficult to assume these effects are still detectable after such a long engraftment period without proof of biological activity/ functional availability.

Author Response

We sincerely thank the reviewer for the constructive criticism. Please see the attachment.

Round 2

Reviewer 2 Report

All my comments were adequately addressed